# Once-Weekly Semaglutide Induces an Early Improvement in Body Composition in Patients with Type 2 Diabetes: A 26-Week Prospective Real-Life Study

**DOI:** 10.3390/nu14122414

**Published:** 2022-06-10

**Authors:** Sara Volpe, Giuseppe Lisco, Davide Racaniello, Margherita Fanelli, Valentina Colaianni, Alfredo Vozza, Vincenzo Triggiani, Carlo Sabbà, Cosimo Tortorella, Giovanni De Pergola, Giuseppina Piazzolla

**Affiliations:** 1Interdisciplinary Department of Medicine, School of Medicine, University of Bari “Aldo Moro”, Piazza Giulio Cesare 11, 70124 Bari, Italy; sarafox77@libero.it (S.V.); g.lisco84@gmail.com (G.L.); racaniello.davide@gmail.com (D.R.); margherita.fanelli@uniba.it (M.F.); v.colaianni1@studenti.uniba.it (V.C.); alfredovozza@live.it (A.V.); vincenzo.triggiani@uniba.it (V.T.); carlo.sabba@uniba.it (C.S.); cosimo.tortorella@gmail.com (C.T.); 2Unit of Internal Medicine and Geriatrics, National Institute of Gastroenterology, “Saverio de Bellis” Research Hospital, Castellana Grotte, Via Turi 27, 70013 Bari, Italy; giovanni.depergola@irccsdebellis.it

**Keywords:** type 2 diabetes, obesity, glucagon-like peptide-1 receptor agonist, Semaglutide, body composition, real-life study

## Abstract

Background: Body weight (BW) loss is an essential therapeutic goal in type 2 diabetes (T2D). Glucagon-like peptide-1 receptor agonists are effective in reducing BW, but their effect on body composition has not yet been fully explored. The study aim was to assess the impact of Semaglutide on body composition in patients with T2D. Methods: Forty patients with T2D were treated with subcutaneous Semaglutide and evaluated at the baseline (T0) and after three (T3) and six (T6) months. Body composition was assessed by a phase-sensitive bioimpedance analyzer. Visceral adipose tissue (VAT) thickness was also measured with an ultrasonographic method (US-VAT). Anthropometric variables, muscular strength, and laboratory tests were analyzed and compared. Results: A significant decrease in VAT, the fat mass index (FMI), and BW loss was observed at all observation times. US-VAT, the skeletal mass index (SMI), the fat-free mass index (FFMI), waist circumferences, and glycated hemoglobin had lessened after three months and remained stable at T6. No variations in muscle strength, the muscle quality index, and body water were found. Discussion: In a real-life setting, Semaglutide provided significant weight loss mainly due to a reduction in the FMI and VAT, with non-clinically relevant changes in the SMI, the FFMI, and muscle strength. Most importantly, the results were obtained after three months of treatment and persisted thereafter.

## 1. Introduction

Type 2 diabetes (T2D) is a metabolic disorder characterized by impaired glucose homeostasis and insulin resistance. It is commonly linked to being overweight/obesity, metabolic syndrome, arterial hypertension, and cardiovascular (CV) and chronic renal diseases [1]. The paradigm of T2D management has changed over the last few years thanks to modern therapeutic opportunities that allow a better control of the glucose metabolism and diabetes-related comorbidities and complications [2]. In particular, glucagon-like peptide-1 receptor agonists (GLP-1RAs) have been shown to improve CV outcomes in T2D [3]. They reduce hyperglycemia by increasing insulin and decreasing glucagon secretion in a glucose-dependent manner, also providing significant weight loss by reducing the appetite and food consumption [4]. BW management is an essential therapeutic goal in T2D; GLP-1RAs are effective in promoting and maintaining weight loss. However, to be healthy, body weight loss should be driven by a clinically relevant loss of fat mass (FM) and visceral adipose tissue (VAT) whilst preserving lean mass, skeletal muscle mass (SMM), and strength.

VAT reduction is accompanied by a decrease in circulating adipokines and cytokines, implicated in low-grade inflammation, insulin resistance, and CV risk [5]. Under insulin resistance conditions, VAT releases considerable amounts of free fatty acids and cytokines into the portal vein circulation, inducing other concomitant conditions in T2D including hepatic insulin resistance and steatosis, hyperglycemia, and dyslipidemia [6]. On the other hand, subcutaneous adipose tissue (SAT) acts as a metabolic sink for additional lipid storage and serves as a barrier against dermal infections and external physical stress agents as well as preventing heat loss [7]. SAT is characterized by more numerous small adipocytes; it is more insulin-sensitive than VAT due to a higher expression of insulin receptor substrate (IRS)-1 [8]. Therefore, the insulin-mediated antilipolytic effect is more robust in SAT than VAT; this effect leads to protection from the excessive bloodstream spilling of free fatty acids.

Skeletal muscle plays a key role in regulating insulin sensitivity and glucose homeostasis as it is the most important site for the disposal of ingested glucose in healthy individuals [9]. Skeletal muscle is responsible for the uptake of around two-thirds of ingested glucose via an insulin-mediated mechanism elicited by a post-prandial glycemic excursion [10]. Sarcopenia—a progressive, generalized, and accelerated loss of skeletal muscle and function—contributes to the onset or exacerbation of T2D, especially in the elderly [11]. Muscle-mass loss synergistically accompanies the accumulation of fat mass in T2D, resulting in so-called “sarcopenic obesity”, a medical condition predisposed to adverse events such as accidental falls, hospitalizations, and frailty [12]. Preserving lean mass, namely fat-free mass (FFM), appears to be a therapeutic goal in T2D as it affects the metabolic aging of patients.

Semaglutide subcutaneously administered once-weekly has been demonstrated to significantly improve glucose control and reduce body weight; it is one of the most potent GLP-1RAs approved for the treatment of T2D and related comorbidities [13,14]. It provides greater weight loss compared with other GLP-1RAs of up to 6 kg over 6 months of treatment [15]. BW loss is usually independent of the baseline BMI and partially independent to gastrointestinal adverse events [14,16]. In addition, Semaglutide significantly reduces composite cardiovascular outcomes in a high-risk population, as demonstrated by the results of SUSTAIN-6 [17]. Other than reducing body weight, the effect of Semaglutide on body composition in T2D has not been completely elucidated in a real-life setting. The aim of this study was to assess the impact of subcutaneous Semaglutide on body composition and muscular strength in T2D outpatients.

## 2. Materials and Methods

### 2.1. Study Design and Institution

This 26-week prospective, single-arm, open-label, real-life study was carried out in the Metabolic Disorders Outpatients Clinic of the Department of Internal Medicine at the University of Bari, Italy.

### 2.2. Ethics

The study was conducted in accordance with the general ethical principles for medical research involving human subjects of the Declaration of Helsinki [18]. The study protocol was formally approved by the Clinical Investigation Ethics Committee of the University of Bari (ID: PZZ_DM2 2020, number 6468, version 2, 14 September 2020).

### 2.3. Inclusion Criteria

The inclusion criteria were an established diagnosis of T2D; age > 18 years; a stable eGFR > 15 mL/min/1.73 m^2^; eligibility for GLP-1RA intensification according to current recommendations and guidelines; and uncontrolled T2D, i.e., glycated hemoglobin (HbA_1c_) > 7% whilst on oral antihyperglycemic medications independent of the baseline HbA_1c_ or individualized glycemic targets in cases with established CV diseases or at a high CV risk and the need to minimize weight gain or promote weight loss [19].

### 2.4. Exclusion Criteria

These included other forms of diabetes mellitus; pregnancy or lactation; inadequate compliance with or contraindications to GLP-1RAs; patients who were prescribed antihyperglycemic medications affecting body composition other than metformin (i.e., GLP-1RAs, sodium-glucose co-transporter-2 inhibitors) prior to the study enrollment; implantable electronic devices (cardioverter defibrillators or pacemakers) as indicated by the manufacturer of the segmental multifrequency bioelectrical impedance analysis (SMF-BIA) system that we used for the assessment of body composition; and an inadequate ability to comply with the follow-up or to provide informed consent.

### 2.5. Study Protocol

Eligible patients were fully informed about the study purposes and gave written consent to participate. The overall study period was 26 weeks. Follow-up visits were carried out quarterly during three scheduled appointments at T0, T3, and T6.

A complete medical history was collected and a physical examination was performed during each follow-up visit. The clinical and anthropometrical parameters analyzed were the office arterial pressure, heart rate, BW, waist circumference (WC), and body mass index (BMI). The laboratory tests included a complete total blood count (Advia 2120i, Siemens Healthineers, Erlangen, Germany), fasting plasma glycemia (FPG, enzymatic method, Dimension Vista system, Siemens Healthineers, Germany), HbA_1c_ (high-performance liquid chromatography, Variant II, Bio-Rad Laboratories, Hercules, CA, USA), serum creatinine (enzymatic method, Dimension Vista system, Siemens Healthineers, Erlangen, Germany) with an estimated glomerular filtration rate (eGFR), fasting plasma insulin, and C-peptide (chemiluminescence immunoassay, Immulite, Siemens Healthinners, Erlangen, Germany). The homeostasis model assessment for insulin resistance (HOMA-IR) index was calculated as (fasting insulin × fasting glucose)/405 (normal range 0.23–2.5).

The body composition was studied by the SMF-BIA [20,21]. A non-invasive analysis of the body composition was performed by a phase-sensitive, octopolar SMF-BIA (Seca mBCA 525; Seca GmbH & Co., KG, Hamburg, Germany) operating with Seca Analytics 115 software. In accordance with the standardized method, the measurements were obtained with patients in a supine position with each leg at an angle of 45° and each arm at an angle of 30° from the trunk. Patients fasted for 8 h and rested for at least 8 h. Body impedance was measured using an alternating current at 100 µA with frequencies ranging from 1–500 kHz; PhA, expressed in degrees, was performed at the single frequency of 50 kHz according to international standards. The raw data (Rz, Xc) were processed by Seca Analytics 115 software to obtain the following values: phase angle (PhA), total body water (TBW), extracellular water (ECW), skeletal muscle mass (SMM), skeletal muscle index (SMI) [22], fat mass index (FMI), fat-free mass index (FFMI), and VAT.

The hand grip (HG) strength test was performed using a manual hydraulic dynamometer (Lafayette Instrument, USA) to assess the muscle strength in both hands. Patients were seated with their arms flexed at 90° and forearms in a neutral position holding a dynamometer. Three separate repeat measurements were recorded for each hand and the best value was selected for the analysis [23]. To more accurately assess the muscle quality, an index that correlated the skeletal muscle mass with the functional data expressed by the HG—namely, the muscle quality index (MQI)—was also calculated by dividing the HG strength value by the SMM (kg/kg).

VAT thickness was measured with an ultrasonographic method (US-VAT) using a General Electrics Logiq E9 Ecograph (GE Healthcare, Milwaukee, WI, USA) according to a previously validated and less invasive method [24]. The US-VAT was evaluated by placing a convex US probe (1–6 MHz) on the abdominal median line around the umbilical scar to visualize the abdominal aorta before the bifurcation, which was measured as the distance between the inner line of the recti muscles and the abdominal aorta anterior wall.

Once-weekly Semaglutide (qw) was administered in accordance with the general recommendations and clinical guidelines. As per good clinical practice, a trial dose of 0.25 mg qw was administered at the time of the first visit. If well-tolerated, the Semaglutide prescription was confirmed as 0.25 mg qw for three consecutive weeks and then raised to 0.5 mg qw. After 3 months of treatment, the Semaglutide was further intensified to 1 mg qw in 2 patients to reinforce the extra glycemic effects (such as promoting a body weight reduction) when clinically indicated. All participants were taking a metformin therapy at the maximum tolerated dose and Semaglutide was prescribed according to the guidelines as an add-on to the metformin or to replace other antihyperglycemic agents such as glinides, sulfonylureas, and dipeptidyl peptidase type IV inhibitors.

#### Study Outcomes

The primary outcome was a change in the VAT, FMI, SMI, FFMI, BW, BMI, and WC from the baseline to T3 and T6. A weight loss of at least 5% of body weight from the baseline to 3 and 6 months was considered to be clinically relevant [25].

The secondary outcome was a change in the HOMA-IR index, FPG, and HbA_1c_ from the baseline to T3 and T6.

### 2.6. Study Participants

A total of 180 patients with T2D (mean age 64.9 ± 10.8 years; men 60.2%) who attended our clinic and who were suitable for an intensification of antihyperglycemic therapy due to poor glycemic control, background cardiovascular and renal conditions, and weight excess were screened for eligibility from 1 October 2020 to 31 March 2021. Among them, 53 (29.4%) patients who were suitable and willing to receive once-weekly subcutaneously administered Semaglutide were included in the study. According to the pre-specified study protocol, thirteen patients were excluded: one due to statistical concerns (outlier), one due to a systematic bias in the bioimpedance output, and eleven because of missing data.

### 2.7. Statistical Analysis

The descriptive statistics were represented as the mean, standard deviation, median, minimum, and maximum values as well as the frequency and percentage. Changes in the variables over the follow-up (i.e., T0, T3, and T6) were analyzed by mixed models for repeated measures; the means were estimated by the least squares’ method. Pearson correlations for the association among variables were calculated. The statistical analysis was performed using SAS software 9.4 (SAS Institute Inc., Cary, NC, USA).

## 3. Results

### 3.1. Descriptive Statistics

The baseline characteristics of the study participants are shown in Table 1. Patients were almost equally distributed with regard to gender (21 men and 19 women), and showed anthropometric parameters typical of a dysmetabolic phenotype such as high average values of BMI and WC.

As attested by the HOMA-IR index values, a significant degree of insulin resistance was also found (Table 1). More precisely, 31 participants (78%) had a HOMA-IR index > 2.5 whereas the remaining 9 did not meet the insulin resistance criterion.

Class II obesity (BMI 35–40 kg/m^2^) was identified in 16 out of 40 (40%) patients; 27.5% had class III obesity (BMI > 40 kg/m^2^, up to 60.7 kg/m^2^) and 32.5% had a BMI between 28.1 (the lowest recorded value) and 35 kg/m^2^. None had normal WC values (< 80 and < 94 cm for women and men, respectively).

According to the novel BMI-dependent SMI cut-offs [26], 6 out of 19 (31.6%) women and 3 out of 21 (14.3%) men met the criterion of sarcopenic obesity. A wide variability in diabetes duration (0–20 years) and glycemic control (HbA_1c_ 33–128 mmol/mol) was noted. Arterial hypertension and hypercholesterolemia were the most frequently diagnosed comorbidities.

At the end of the trial, 95% of patients were on Semaglutide 0.5 mg qw; in the remaining 5%, the Semaglutide dosage was raised to 1 mg qw, specifically to promote body weight loss.

### 3.2. Body Weight and Composition

A positive correlation was found between the measurements of VAT with the SFM-BIA and US (r = 0.61548; *p* < 0.001), suggesting that both methods offered a reliable estimate of the amount of visceral adipose tissue. A significant and progressive decrease in VAT was observed throughout the study period whereas the reduction in US-VAT was significant at T3, but not at T6 (Figure 1 and Table 2).

The changes in the FMI, SMI, and FFMI are illustrated in Figure 2 and Table 2. The FMI significantly decreased at all observation times whereas the SMI and FFMI had lessened after 3 months of therapy and remained stable at the 26th week. As shown in Table 2, the HG strength test did not reveal any decline in muscle strength over time (F = 0.19; *p* = 0.82), nor did the MQI (F = 2.45; *p* = 0.1). Moreover, the HG strength test was found to be positively associated with SMM (r = 0.38034; *p* = 0.02). 

Both total and extracellular body water remained unchanged over time (TBW: F = 0.38; *p* = 0.69 and ECW: F = 0.42; *p* = 0.66) and no significant variations in PhA were observed after 26 weeks of treatment (PhA: F = 2.3; *p* = 0.11) (Table 2).

As shown in Table 2, a significant reduction in the BW, BMI, and WC was found after 3 months of treatment compared with the baseline and the decrease was confirmed after 26 weeks. The BW continued to decrease at week 26 and was significantly lower at T6 compared with T3. Weight loss was observed in all study participants. In particular, 71.4% of patients obtained a BW reduction of 5% or more after three months of treatment; these were classified as responder patients. The percentage of responders rose to 77.5% after six months of therapy (Figure 3).

Crossing the SMF-BIA measures with the anthropometric data, 10 out of 38 (26.3%) patients with complete SMF-BIA measurements exhibited a loss in FFM (kg) greater than 40% of the total amount of weight loss after 6 months of therapy; only 2 patients lost more than 40% of SMM (kg). In 32 out of 38 patients (84.2%), the weight loss was prevalently driven by a fat mass decline by more than 40%.

### 3.3. Glycemic Control

The HbA_1c_ values significantly improved after three months of treatment and remained stable at T6 whereas FPG declined up to T6 (Table 2).

Fasting serum insulin and the HOMA-IR index were both found to be significantly lower at week 26 than the baseline (Table 2). Conversely, the C-peptide (F = 0.47; *p* = 0.63), serum creatinine (F = 1.52; *p* = 0.23), and eGFR (F = 0.61; *p* = 0.55) values remained unchanged throughout the study period (Table 2).

## 4. Discussion

T2D and visceral obesity are currently considered to be a global health emergency, mainly due to their rising prevalence. The accumulation of VAT is a well-known risk factor underlying the development of insulin resistance, T2D, and CVD [27]. In addition, T2D patients are prone to lose muscle mass more rapidly than non-diabetic people, thus becoming more susceptible to insulin resistance than the general population. Promoting long-term weight loss is an important, but challenging, therapeutic goal in patients with T2D, as recently suggested [28]. Drastic low-calorie diets and bariatric surgery are also highly effective interventions, inducing a significant reduction in adipose tissue [29,30]. However, it should be considered that the beneficial effects resulting from a 5–10% weight loss can be nullified if there is a concomitant exaggerated loss of lean mass. In the latter case, patients are prone to losing muscle strength and physical efficiency as well as developing insulin resistance; these are all conditions that are ultimately predisposed to weight regain and worse long-term glycemic control.

The assessment of excess weight and adipose tissue distribution using the BMI and waist circumference parameters is a quick and inexpensive method to summarize the cardiometabolic burden of T2D patients and to fully assess the specific impact of any intervention on body weight and fat mass distribution. Nevertheless, this method provides no additional information about the health status of skeletal muscle and lean mass although both of these play a crucial role in insulin sensitivity and have a cardiovascular benefit [31]. On this basis, the assessment of body composition is a valuable and easy-to-use tool for obtaining better information at the baseline and during follow-up.

The impact of GLP-1RAs on glucose control and weight loss in T2D and overweight or obese patients is well-known, but the effect on body composition needs to be better elucidated, especially in a real-life setting [32,33]. In this prospective real-life study, we evaluated the impact of Semaglutide on body composition using non-invasive, highly reproducible, and inexpensive methods. The SMF-BIA estimates body composition with extreme precision compared with reference methods such as sodium bromide, deuterium dilution, or dual-energy X-ray absorptiometry [34]. The HG strength test and the US measurements of visceral adipose tissue thickness are other manageable and accurate procedures to assess muscle strength and visceral abdominal fat distribution, respectively.

Our data demonstrated that a treatment with once-weekly subcutaneous Semaglutide ensured an adequate glucometabolic control whilst improving insulin resistance and promoting weight loss (9.9 kg after 6 months of treatment) without compromising body composition. The therapy-induced significant reductions in BMI and WC were driven by a decline in both VAT and fat mass over time, which are potentially predisposed to a decrease in future CV risk. The concomitant reduction in lean mass and skeletal muscle mass was also expected, but it could be considered to be clinically irrelevant. When BW loss occurs with a decline in lean and skeletal mass not exceeding 40% of the total BW loss, this loss can be considered to be healthy. In our study, only 2 out of 40 patients exhibited more than 40% of SMM loss without a relevant decline in skeletal muscle strength whereas 84% lost weight due to a fat mass decline of more than 40%.

Neutral effects were observed for the body water content and phase angle, an index of conductive tissue (muscle mass) integrity expressing the goodness of weight loss in terms of body composition. To reinforce the above-mentioned findings, it should be stressed that muscle strength, assessed by the HG strength test, was also preserved, thus suggesting that body weight loss did not affect the skeletal muscle performance.

Interestingly, most of these effects had a rapid onset as they were already evident after three months of therapy and persisted or further improved over time (e.g., FPG, weight loss, VAT, and FMI reduction). Conversely, the improvement in insulin sensitivity occurred more slowly as the reduction in the HOMA-IR index was found to be relevant only after six months of treatment. Therefore, weight loss and improvement in body composition occurred before the amelioration of insulin sensitivity, suggesting that the latter effect might result from a synergistic interaction of the earlier metabolic effects of Semaglutide.

Notably, no specific prescription in terms of daily physical exercise was provided and the study results were achieved without significant changes to background caloric daily expenditure.

To our knowledge, this is the first prospective real-life study to comprehensively assess the role of Semaglutide, a GLP-1RA, on body composition in a cohort of T2D patients over a 26-week follow-up. A sample size determination to control for type II errors was not calculated due to the lack of previously published results in the field. The other study limitations were: (i) the lack of nutritional information before and during follow-up, even if no specific adjustment of caloric intake was prescribed to study participants; (ii) the assessment of muscle strength of the arms, but not the legs; and (iii) the lack of a control group (i.e., patients not receiving Semaglutide), which was not included given that the study aimed to specifically assess the “quality” of Semaglutide-induced weight loss.

## 5. Conclusions

In a real-life setting, Semaglutide provided significant weight loss, predominantly due to a reduction in fat mass and VAT with a mild decline in the FFMI and skeletal muscle mass not associated with a loss of muscle strength. The changes occurred rapidly and without relevant modifications to lifestyle and nutritional behaviors.

The use of GLP-1RAs in a larger population of patients with T2D would be desirable in order to promote weight loss whilst improving body composition along with a specific nutritional, physical, and health education plan aimed at preserving skeletal muscle mass.

A bioelectrical impedance analysis can be considered to be a simple-to-use and cost-effective tool to quickly assess body composition for a better ascertainment of weight loss quality in T2D patients. Based on our experience from a university tertiary care center, its use should be boosted in diabetes care as well as in other settings.

## Figures and Tables

**Figure 1 nutrients-14-02414-f001:**
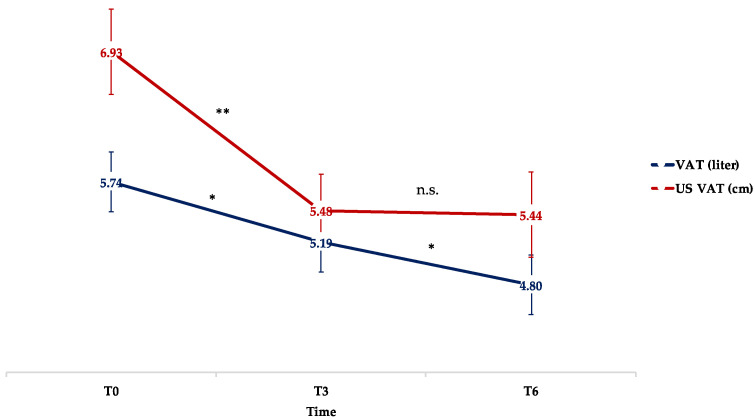
Mean changes in VAT (L) and US-VAT (cm) throughout the study period. Means are reported as least squared means estimated by mixed model analysis. * *p* < 0.05; ** *p* < 0.01; n.s.: not significant; VAT: visceral adipose tissue; US-VAT: ultrasonographic visceral adipose tissue.

**Figure 2 nutrients-14-02414-f002:**
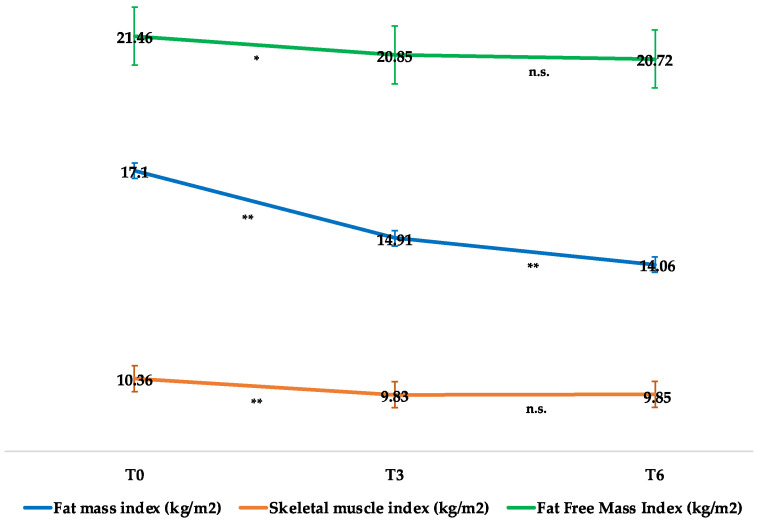
Mean changes in fat mass index, skeletal muscle index, and fat-free mass index throughout the study period. Means are reported as least squared means estimated by mixed model analysis. * *p* < 0.05; ** *p* < 0.01; n.s.: not significant.

**Figure 3 nutrients-14-02414-f003:**
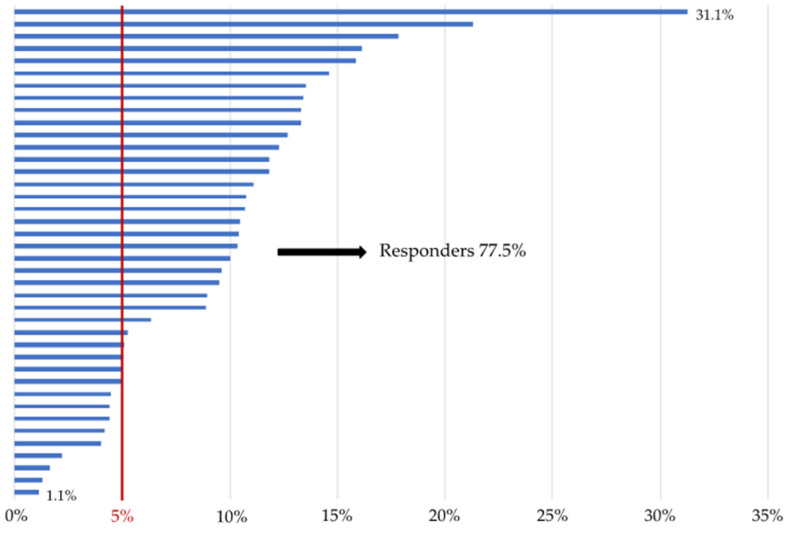
Graphical representation of the percentage of weight loss for each participant. (N= 40) from baseline to week 26. Responders were defined as participants achieving a body weight loss of 5% or more after six months of treatment.

**Table 1 nutrients-14-02414-t001:** Baseline characteristics of study participants.

Baseline Characteristics	Mean (s.d.)	Median (min; max)
Body mass index (kg/m^2^)	38.8 (7.7)	36.9 (28.1; 60.7)
Waist circumference (cm)	123.4 (14.4)	120 (101; 161)
Fasting glycemia (mg/dL)	129.9 (36.6)	121.5 (87; 221)
Glycated hemoglobin (mmol/mol)	52.9 (21.6)	44.5 (33; 128)
Serum creatinine (mg/dL)	0.9 (0.2)	0.85 (0.40; 1.65)
Glomerular filtration rate (mL/min/1.73 m^2^)	87.8 (22.9)	88 (41; 153)
Fasting serum insulin (mUI/L)	22.6 (15.2)	20.1 (3.8; 75)
Fasting serum C-peptide (ng/mL)	3.7 (1.5)	3.4 (1.3; 8.1)
HOMA-IR index	6.9 (4.7)	5.3 (1.6; 21.8)
Visceral adipose tissue (L)	5.7 (2.9)	5.1 (2.3; 14.9)
Fat mass index (kg/m^2^)	17.1 (6.3)	15.7 (6; 36.6)
Fat-free mass index (kg/m^2^)	21.4 (3.1)	21.7 (15.7; 29.8)
Skeletal muscle mass (kg) Skeletal muscle index (kg/m^2^) Hand grip (kg) MQI (kg/kg)	28.2 (6.2) 10.4 (1.8) 32.5 (9.8) 1.1 (0.6)	28.3 (15.9; 42.1) 10.7 (6.8; 15.10) 32 (15; 60) 1.1 (0.5; 3.15)
Total body water (L)	42.9 (9.1)	42.7 (24; 62.6)
Extracellular body water (L)	19.4 (3.7)	18.6 (12.7; 28)
Extracellular body water to total body water ratio	45.6 (2.6)	44.8 (40.9; 53)
Phase angle (°)	5.8 (0.8)	5.7 (3.9; 7.5)
US-VAT (cm)	6.9 (2.4)	6.5 (3; 13)

Descriptive statistics are presented as means, standard deviation, and median, minimum, and maximum values. HOMA-IR: homeostatic model assessment for insulin resistance; MQI: muscle quality index; US-VAT: ultrasonographic visceral adipose tissue.

**Table 2 nutrients-14-02414-t002:** Estimated mean changes in anthropometric, serologic, bioimpedance, and ultrasonographic parameters over the study period (T3 and T6).

Parameters	Time		
	T0	Variation at T3	Variation at T6
Body weight (kg)	103.96 ± 3.03	−7.83 ± 0.72 **	−9.89 ± 0.99 ** #
Body mass index (kg/m^2^)	38.81 ± 1.18	−3.05 ± 0.30 **	−3.36 ± 0.42 **
Waist circumference (cm)	123.53 ± 2.24	−6.32 ± 1.12 **	−7.31 ± 1.15 **
Fasting glycemia (mg/dL)	129.95 ± 5.71	−15.57 ± 4.41 **	−23.56 ± 4.45 ** ##
HbA_1c_ (mmol/mol)	52.86 ± 3.36	−10.72 ± 2.80 **	−11.16 ± 2.99 **
Serum creatinine (mg/dL)	0.88 ± 0.04	0.03 ± 0.03	0.01 ± 0.03
eGFR (mL/min/1.73 m^2^)	87.85 ± 3.57	0.85 ± 2.55	−2.02 ± 2.52
Fasting serum insulin (mUI/L)	22.59 ± 2.46	−0.76 ± 3.17	−5.22 ± 2.16 *
Fasting serum C-peptide (ng/mL)	3.72 ± 0.24	0.08 ± 0.29	−0.13 ± 0.27
HOMA-IR index	6.88 ± 0.77	−1.22 ± 1.06	−2.62 ± 0.79 **
Visceral adipose tissue (L)	5.74 ± 0.45	−0.55 ± 0.27 *	−0.95 ± 0.24 * #
Fat mass index (kg/m^2^)	17.10 ± 0.99	−2.19 ± 0.46 **	−3.04 ± 0.43 ** #
Fat-free mass index (kg/m^2^)	21.45 ± 0.47	−0.61 ± 0.24 *	−0.74 ± 0.17 **
Skeletal muscle mass (kg)	28.16 ± 0.98	−1.31 ± 0.37 **	−1.53 ± 0.36 **
Skeletal muscle index (kg/m^2^)	10.36 ± 0.27	−0.52 ± 0.14 **	−0.51 ± 0.14 **
HG (kg)	32.49 ± 1.64	0.49 ± 1.75	0.76 ± 1.26
MQI (kg/kg)	1.06 ± 0.09	0.16 ± 0.09	0.17 ± 0.08
Total body water (L)	42.95 ± 1.39	−0.20 ± 0.79	−0.87 ± 1.13
Extracellular body water (L)	19.45 ± 0.56	−0.10 ± 0.32	−0.41 ± 0.48
ECW to TBW ratio	45.6 ± 0.41	−0.24 ± 0.36	0.00 ± 0.35
Phase angle (°)	5.76 ± 0.12	−0.15 ± 0.10	−0.21 ± 0.10
US-VAT (cm)	6.93 ± 0.39	−1.45 ± 0.23 **	−1.49 ± 0.33 **

Models for each parameter are expressed as mean ± S.E. at T0 and mean variation ± S.E. at T3 and T6. HbA_1c_: glycated hemoglobin; eGFR: estimated glomerular filtration rate; HOMA-IR: homeostatic model assessment for insulin resistance; HG: hand grip; MQI: muscle quality index; ECW: extracellular body water; TBW: total body water; US-VAT: ultrasonographic visceral adipose tissue. Variation versus T0: * *p* < 0.05; ** *p* < 0.01. Variation versus T3: # *p* < 0.05; ## *p* < 0.01.

## Data Availability

Data supporting the results are available on reasonable request to the corresponding author.

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
