# Peer review of "Once-Weekly Semaglutide Induces an Early Improvement in Body Composition in Patients with Type 2 Diabetes: A 26-Week Prospective Real-Life Study"

_nutrients, 2022, doi:10.3390/nu14122414_

Round 1

Reviewer 1 Report

This study focused on the impact of semaglutide on body composition in patients with T2D in a real-life setting for the first time. It was of great scientific value. The experimental design was reasonable, and the results could be well explained and effectively supported the conclusion. Some minor comments are as below.  

1. Please delete the full stop of the title.

2. Please follow the Abstract format of the journal Nutrients.

3. The abbreviation “GLP1-RAs” should not occur in the Keywords.

4. Please add more details and recent advances about semaglutide in the section of Introduction.

5. Line 112-115, please specify the methods.

6. For Fig.1 and Fig.2, please add the error bar for each data point.

Author Response

The authors thank the reviewers for their helpful comments and suggestions serving to improve the manuscript quality. Herein, a point-to-point response has been provided. All changes are in red color.

Please delete the full stop of the title.

The full stop has been deleted.

Please follow the Abstract format of the journal Nutrients.

The abstract is now formatted according to the journal recommendations.

The abbreviation “GLP1-RAs” should not occur in the Keywords.

“GLP1-RAs” has been deleted from the keywords.

Please add more details and recent advances about Semaglutide in the section of Introduction.

Other information has been added in the introduction. Please check the Introduction text for details

Line 112-115, please specify the methods.

Methods have been specified.

For Fig.1 and Fig.2, please add the error bar for each data point.

Both figures have been updated according to the suggestion.

Reviewer 2 Report

Volpe and colleagues investigated the potential for Semaglutide to improve body composition of patients with T2DM in a tertiary hospital in Italy. Overall, the methods employed in the study are appropriate and the findings are interesting and may also have clinical relevance. The manuscript is also nicely written.

Methods

1)     It is not clear how the participants who met the eligibility criteria were selected. How many were invited, and what was the response rate? Did those included in the study have similar characteristics as those not included in terms of age, sex and other important characteristics?

2)      Line 154: “weight loss of at least 5% of body weight from baseline to 3 and 154 6 months was considered clinically relevant.” Why 5%?

3)      Any justification for using mean instead of median, given that the vast majority of biochemical measurements do not follow normal distribution?

Discussion

4)      Was the sample size appropriate for the analysis? Is the study not prone to type II error, given the small sample size? This should be discussed as a potential limitation.

5)      Is the apparent improvement in body composition not also attributed to changes in lifestyle following diagnosis of T2DM? Since no control group (those not receiving Semaglutide) was analysed in the study, the authors should interpret the findings with caution and acknowledge this limitation as well.

6)      The participants were selected from one health facility (likely a tertiary hospital). Can the findings from the study be generalised to other settings?

Author Response

The authors thank the reviewers for their helpful comments and suggestions serving to improve the manuscript quality. Herein, a point-to-point response has been provided. All changes are in red color.

It is not clear how the participants who met the eligibility criteria were selected. How many were invited, and what was the response rate? Did those included in the study have similar characteristics as those not included in terms of age, sex and other important characteristics?

We are grateful to the reviewer for this important clarification request. Accordingly, in section 2.6 -Study participants- we have more fully described the characteristics of the population screened and then enrolled in the study. Patients attending our center for diabetes care were consecutively selected and screened for eligibility for Semaglutide based upon background clinical characteristics and personal preference. According to these criteria, 29.4% of patients were assigned to subcutaneous Semaglutide. They were actually slightly younger and had a more even gender distribution  than the other screened patients. Details have been provided in the text (see section 2.6).

Line 154: “weight loss of at least 5% of body weight from baseline to 3 and 6 months was considered clinically relevant.” Why 5%?

Evidence suggests that achieving a  weight loss of 5% or more in 3 to 6 months effectively reduces cardiovascular risk in T2D. Also, guidelines suggest this cut-off point as a primary therapeutic goal. In the present version of the manuscript, we have added a bibliographic reference to support this evidence.

Any justification for using mean instead of median, given that the vast majority of biochemical measurements do not follow normal distribution?

As suggested, the median values, as well as the minimum and maximum values of the analyzed parameters, have been added. Please see Table 1 for details.

Was the sample size appropriate for the analysis? Is the study not prone to type II error, given the small sample size? This should be discussed as a potential limitation.

As suggested, in the current version of the manuscript we discuss this point as a potential limitation,  specifying that “Sample size determination to control for type II error was not calculated due to the absence of previously published results in the field”.

Is the apparent improvement in body composition not also attributed to changes in lifestyle following diagnosis of T2DM? Since no control group (those not receiving Semaglutide) was analysed in the study, the authors should interpret the findings with caution and acknowledge this limitation as well.

With reference to the first point, it should be considered that only 5/40 patients were first diagnosed with T2D during the study. Moreover, lifestyle modifications are an essential part of T2D treatment, but most patients fail to follow recommendations. Pharmacological intensification is necessary to ameliorate glycemic control. Herein, patients did not receive specific prescriptions for physical exercise and diet. Therefore, we believe that the results are reasonably attributable to pharmacological intervention.

As suggested by the reviewer, the lack of a control group is listed among the study limitations. The inclusion of a control group was not deemed necessary because Semaglutide has already been shown to provide significant weight loss, hence the study aimed to specifically assess the  “quality” of this weight loss. Moreover, this was a real-life study in which participants were assigned to receive a GLP-1RA due to background characteristics. Therefore, generating a control group (i.e., placebo) could have raised relevant ethical concerns. For this reason, our group is already conducting a comparative study focusing on the effects on body composition of several new anti-diabetic drugs (i.e. SGLT2i and Dulaglutide).

The participants were selected from one health facility (likely a tertiary hospital). Can the findings from the study be generalized to other settings?

The study of body composition could improve the quality of care of T2D patients as it provides additional parameters in addition to the anthropometric variables usually used (BMI and waist circumference). The bioelectrical impedance analysis is an easy-to-use and cost-effective tool for quickly providing real-time body composition results. Thanks to this comment, we have added in the conclusions that, based on our experience as a university tertiary care center, we could suggest the use of this tool also in other care settings.

Reviewer 3 Report

Dear author! For research, a group of subjects, physical methods for evaluating results, and statistical data processing were carefully selected. Semaglutide was chosen as a drug for the treatment of type 2 diabetes. The only advantage of the drug is the presence of two biguanidine groups. A similar chemical group exists in the metformin molecule, which is a nitric oxide (NO) donor. The drug Semaglutide has a high cost, side effects due to the induction of a thyroid tumor, inflammation in the pancreas and gallbladder. In my opinion, detailed studies of Semaglutide on NO donor properties are necessary and only then should it be introduced into medical practice. In the presented work, metformin, which has a long-term positive reputation, was used to enhance the effect. In model studies, metformin also demonstrated antitumor activity. I believe that the NO donor properties of Semaglutide should be considered in the editor for subsequent publication of the materials.

Author Response

The authors thank the reviewers for their helpful comments and suggestions serving to improve the manuscript quality. Herein, a point-to-point response has been provided. All changes are in red color.

We thank this reviewer  for the comment. Efficacy/effectiveness and safety of Semaglutide have been demonstrated in clinical trials. According to the results of the SUSTAIN program, the rates of discontinuation and requirement for rescue therapy, both considered the main safety points, ranged from 6 to 20% for Semaglutide 0.5 mg and 5 to 22% for Semaglutide 1 mg, and 0.7 to 17% for Semaglutide 0.5 mg and 1 to 18% for Semaglutide 1 mg, respectively (Smits MM et al. Front Endocrinol 2021). In less than one-third of the cases, discontinuation was attributable to adverse events, that were usually mild or moderate. Acute pancreatitis is more frequently observed in people with diabetes than in healthy individuals (Gonzalez Perez A et al. Diabetes Care 2010). Semaglutide and other GLP-1RAs do not seem to increase the risk of acute pancreatitis compared to other medications or placebo (0.3% vs. 0.4%, Fang-Hong Sci et al. Front Pharmacol 2018). Pre-clinical observations suggested that Liraglutide, a long-acting GLP-1RAs, induced c-cell hyperplasia in rodents. This was attributable to direct stimulation of GLP-1 receptors (highly expressed in rodents' parafollicular c-cells), promoting cell growth (Knudsen et al. Endocrinology 2010). As c-cell hyperplasia is a precancerous condition leading to the future development of medullary thyroid cancer in rodents, this phenomenon raised possible concerns for human safety. Regulatory agencies include serial Calcitonin measurements through clinical trials involving the use of GLP-1RAs. However, the incidence rate of medullary thyroid carcinoma is very low, and the causative role of GLP-1RAs has not been confirmed (Chiu WY et al. Exp Diabetes Res. 2012). As a reasonable compromise, despite the lack of evidence, the use of GLP-1RAs (including Semaglutide) should be avoided in "high-risk" patients such as those with Multiple Endocrine Neoplasia type 2 or with an established genetic predisposition to medullary carcinoma (Parks M et al. NEJM 2010). In our study, no patients belonged to these categories. Lastly, thanks for the suggestions concerning NO donor. This issue lies outside the scope of this study, but it might be assessed in other specific trials.

Round 2

Reviewer 3 Report

The work was carried out on a large experimental material, the data are statistically convincing. Wishing the authors to try to justify the results of weight loss by reducing the bioavailability of nitric oxide and submit a new publication on the drug.